# Computational Analysis of the Crystal and Cryo-EM Structures of P-Loop Channels with Drugs

**DOI:** 10.3390/ijms22158143

**Published:** 2021-07-29

**Authors:** Denis B. Tikhonov, Boris S. Zhorov

**Affiliations:** 1Laboratory of Biophysics of Synaptic Processes, Sechenov Institute of Evolutionary Physiology and Biochemistry, Russian Academy of Sciences, St. Petersburg 194223, Russia; zhorov@mcmaster.ca; 2Department of Biochemistry & Biomedical Sciences, McMaster University, Hamilton, ON L8S 4K1, Canada

**Keywords:** ligand-receptor interactions, sequence alignment, 3D alignment, π-bulges, Monte Carlo minimization

## Abstract

The superfamily of P-loop channels includes various potassium channels, voltage-gated sodium and calcium channels, transient receptor potential channels, and ionotropic glutamate receptors. Despite huge structural and functional diversity of the channels, their pore-forming domain has a conserved folding. In the past two decades, scores of atomic-scale structures of P-loop channels with medically important drugs in the inner pore have been published. High structural diversity of these complexes complicates the comparative analysis of these structures. Here we 3D-aligned structures of drug-bound P-loop channels, compared their geometric characteristics, and analyzed the energetics of ligand-channel interactions. In the superimposed structures drugs occupy most of the sterically available space in the inner pore and subunit/repeat interfaces. Cationic groups of some drugs occupy vacant binding sites of permeant ions in the inner pore and selectivity-filter region. Various electroneutral drugs, lipids, and detergent molecules are seen in the interfaces between subunits/repeats. In many structures the drugs strongly interact with lipid and detergent molecules, but physiological relevance of such interactions is unclear. Some eukaryotic sodium and calcium channels have state-dependent or drug-induced π-bulges in the inner helices, which would be difficult to predict. The drug-induced π-bulges may represent a novel mechanism of gating modulation.

## 1. Introduction

Since the breakthrough structure of the KcsA potassium channel was published in 1998 [1], progress in the X-ray crystallography and development of high-resolution cryo-electron microscopy greatly facilitated structural studies of ion channels in different functional states and in complexes with different ligands. Among multiple ion channel structures deposited in the Protein Data Bank [2], few hundred belong to the category of P-loop channels. This very diverse superfamily includes potassium, sodium, and calcium channels, TRP channels, and ionotropic glutamate receptors [3]. Many channels are homo- or hetero-tetramers assembled from subunits. In contrast, the alpha subunit of eukaryotic sodium and calcium channels folds from a single polypeptide chain of four homologous repeat domains. The pore module of P-loop channels has a conserved architecture of the transmembrane region formed by eight transmembrane helices connected by four membrane reentrant P-loops, which harbor the selectivity-filter. The latter divides the ion permeation pathway into the outer pore, which is lined by residues in the C-part of P-loops, and the inner pore, which is lined by the inner helices. Residues at the C-terminal part of the inner helices contribute to the activation gate, which usually opens up in response to membrane depolarization or hyperpolarization (voltage-gated channels) or binding of various ligands (ligand-gated channels). 

Despite generally conserved architecture of the transmembrane part of P-loop channels, the pore modules of individual subfamilies have structural peculiarities. For example, in potassium channels the P-loop of each subunit has a single helix (P1) N-terminal to the selectivity filter, whereas sodium and calcium channels also have a second helix (P2) C-terminal to the selectivity filter. Interfaces (fenestrations) between subunits of potassium channels are rather narrow, but in sodium and calcium channels they are wide, providing a hydrophobic access pathway for some ligands into the inner pore. Mechanisms of channel gating and regulation may also dramatically differ and respective structural changes in the inner helices are also different. In some channels the inner helices have π-helix bulges, which reorient residues vs. analogous channels with regular alpha helices. 

The pore module of P-loop channels is targeted by synthetic drugs and naturally occurring toxins of highly diverse structure. Examples of synthetic ligands range from relatively simple tetraalkylammonium blockers of potassium channels to phenylalkylamines, benzothiazepines, and dihydropyridines that selectively target L-type calcium channels. Some classes of pore-targeting ligands are of high medical importance. Examples include, but are not limited to local anesthetics and anticonvulsants that block voltage-gated sodium channels and antihypertensive blockers of calcium channels. The permanent demand to develop new potent and selective drugs for different P-loop channels stimulates intensive studies in academy and industry. 

The Protein Data Bank contains hundreds of crystal and cryo-EM structures of P-loop channels complexed with various ligands, including drugs, toxins, lipids, and detergent molecules. High structural diversity of these complexes, different orientations of the channels and different numbering of residues complicate comparative analysis of these structures. In this study, we 3D-aligned structures of different P-loop channels, designated residues with universal residue labels, estimated energetics of ligand-channel interactions. Based on these data, we described common and specific features of ligand-channel complexes. 

## 2. Results

### 2.1. Ligand-Binding Sites

For comparison of different structures, we used universal residue labels. A residue label contains subunit/repeat number (1 to 4 for repeats/subunits I to IV), a letter "i", "o", or "p" for the Inner helix, Outer helix and P-loop, respectively, and relative number of the residue in the segment. Sequence alignment of representative channel segments is shown in Figure 1. Figure 2A shows C^α^ tracings of the pore module in 3D-aligned structures of 25 P-loop channels, which are listed in the Appendix A. The RMS deviations of C^α^ atoms in P1 helices (positions p37 to p48) from the Kv1.2/Kv2.1 template is ~1 Å for potassium channels and 2-3 Å for other channels (data not shown). Despite the fact that only C^α^ atoms in four P1 helices were used to minimize the RMS deviations from the Kv1.2/Kv2.1 template, the RMS deviations of C-parts of the outer helices and N-parts of the inner helices are as small as 2–4 Å, demonstrating the common architecture of P-loop channels. Deviations of N-parts of the outer helices and C-parts of the inner helices were much larger due to vastly different backbone conformations at the activation gate region. 

In the 3D-aligned structures, three major ligand-binding regions are seen. The first region is the central cavity (Figure 2B) lined by the inner helices and few residues from the C-terminal part of P1 helices. The second region is fenestrations between one P1 helix and two adjacent inner helices (Figure 2C). In eukaryotic sodium and calcium channels ligands are often seen in the III/IV fenestration. Fluoxetine derivatives occupy an analogous site in the two-pore domain channel TREK2 [4]. Structures of homotetrameric channels, such as NavAb and CavAb, were rotated around the pore axis so that their ligands occurred in the III/IV fenestration. Most ligands bind in the narrow parts of fenestrations (Figure 2C), but some CavAb ligands [5] are rather far from the pore axis. The third ligand-binding region is located at the activation gate, which in the open state can accommodate bulky steroidal molecules (Figure 2D). Small ligands bind in the central cavity or fenestrations. Long molecules can occupy both the central cavity and a fenestration. 

### 2.2. Pore Blockers of Potassium Channels, Prokaryotic Sodium Channels, and GluR Channels

The pore block of potassium channels is exemplified by the structures of KcsA with tetrabutylammonium (PDB ID: 2hvj) [6] and tetradecylammonium (PDB ID: 2w0f) [7] in the central cavity (Figure 3A,B). In these and other KcsA complexes with tetraalkylammonium blockers, the ammonium nitrogen binds at the focus of pore helices. Inner-helix residues I^i15^ and F^i18^, which form tight intersubunit contacts, and residues in positions A^p47^, T^p48,^ and T^p49^ contribute, respectively, ~70 and ~30% to the binding energy of tetrabutylammonium Appendix A. Long hydrophobic chains of tetradecylammonium deeply penetrate into the fenestrations, reaching as far as the outer-helix residue L^o13^
Appendix A. The side chains of F^i18^ remain in tight contact with I^i15^, but reorient to let protrusion of the long decyl chains into the fenestrations.

Block of ionotropic glutamate receptors is exemplified by the NMDAR channel with MK801 (PDB ID: 5un1) [8] and AMPAR channel with IEM-1460 (PDB ID: 6dm0) [9]. In Figure 3C, these structures are superimposed with TBA-bound KcsA (PDB ID: 2hvj) [6] and KcsA with six potassium ions (PDB ID: 3stl) [10]. Geometry P-loops in potassium channels and NMDAR channels are significantly different (Figure 3C). Due to this peculiarity, the ammonium nitrogen of MK-801 is 3 Å closer to the activation gate region in the NMDAR channel as compared to TBA in KcsA (Figure 3C). Position of the MK801 ammonium nitrogen is close to that of potassium ion in the central cavity (site S_5_). The pore-facing residues in positions i15, i17, i18, i19, and i22 contribute to the binding energy of MK801 Appendix A. Among P-loop residues, the major contribution to the ligand-binding energy is provided by N^p49^ whose side chain approaches the ligand ammonium group. In the AMPAR channel with IEM-1460, the major contributions to the ligand-binding energy are provided by glutamines Q^p49^ and Q^p50^. The long chain of IEM-1460 deeply penetrates into the selectivity-filter region and interacts with the backbone atoms of G^p51^, C^p52^, and D^p53^
Appendix A. The ammonium nitrogen atom near the adamantane core of IEM-1460 is as close as 0.3 Å from the nitrogen atom of KcsA-bound TBA, whereas the terminal ammonium nitrogen of IEM-1460 is 0.9 Å from position of potassium ion in site S_3_ of KcsA. The adamantane core of IEM-1460 fits in the central cavity and interacts with I^i18^ in the inner helices. Thus, ligands of dramatically different chemical structures have some common binding determinants in KcsA and iGluR channels.

There are several crystal structures of ligand-bound prokaryotic sodium channels NavAb, NavMs, and NavAb-derived engendered calcium channel CavAb. A DHP derivative efonidipine binds in the central cavity of CavAb and interacts with the P-loop turn and S6 helices (PDB ID: 6juh) [11]. Another DHP derivative, UK59811, is shifted toward the activation-gate region of CavAb (PDB ID: 5klg) [5]. The terminal aromatic ring of UK59811 interacts with V^i22^, which proves the largest contribution to ligand-binding energy, whereas the ligand ammonium group approaches the P-loop turn and interacts the backbone carbonyls, which are not H-bonded to the backbone amide groups Appendix A. 

In the crystal structure of CavAb with Br-verapamil (PDB ID: 5kmh) [5], the entire ligand binds in the inner pore and the ammonium nitrogen approaches the selectivity filter, at the distance of 1.4 Å from position of the ammonium nitrogen in the TBA-bound KcsA. Helices P1 and S6 equally contribute to the binding energy of Br-verapamil, and M^i18^ is the major S6 contributor Appendix A. Crystal structures of CavAb with diltiazem [12] demonstrate a similar pattern of ligand-channel interactions (PDB IDs: 6ke5, 6keb). In the crystal structures of CavAb with DHP antagonists amlodipine (PDB ID: 5kmd), nimodipine (PDB ID: 5kmf), and UK 59811 (PDB ID: 5kls) [5] the ligands bind in the fenestrations, rather far from the pore axis (Figure 2C). In the crystal structure of NavAb with flecainide (PDB ID: 6mvx) [13] the ligand tightly binds near the P-loop turn with the terminal ring approaching the selectivity filter, and therefore P-loop residues contribute to the binding energy much stronger than S6 residues i15 and i18 Appendix A. 

Thus, in the 3D-aligned structures of ligand-bound channels iGluR, CavAb, and NavAb, the ligands fill up the entire central cavity and interact with the P-loop turn (positions p47-p49) and the pore-facing residues in inner helices (positions i15, i18, and i22). The backbone carbonyls at the P-loop turn, which do not accept H-bonds form the mainchain NH groups, attract the ligands’ ammonium groups, providing a significant contribution to the ligand-binding energy. However, positions of the ammonium nitrogen are scattered over the central cavity (blue spheres in Figure 3D). Three overlapping nitrogen atoms in the focus of P-helical dipoles belong to ligands in KcsA (PDB ID: 2hvj and 2w0f) and in the AMPAR channel (PDB ID: 6dm0). We do not see highly specific ligand-channel interactions, which could explain the drug selectivity and structure-activity relationships. 

### 2.3. π-Bulges in P-loop Channels and Their Complexes with Ligands

The above considered channels have somewhat different 3D dispositions of the pore-lining helices, but generally similar orientation of residues in the regular helices that line inner pore and fenestrations (Figure 4A). In contrast, multi-ligand-gated potassium channel GsuK [14] has a π-helix bulge at position i19 (PDB IDs: 4gx0, 4gx1, 4gx5). As a result, orientation of residues downstream position i19 is essentially different as compared to the channels without the bulge (Figure 4A). 

Figure 4B shows deviations of S6 alpha carbons from the KcsA template with regular alpha helices (PDB ID: 1bl8). Unsurprisingly, deviations of C^α^ atoms in another KcsA structure (PDB ID: 2boc) are rather small. In the Kv1.2 structure (PDB ID: 2r9r) deviations are larger due to the open conformation of the activation gate, which is lined by the inner helix. However, the deviations plot is still rather smooth (Figure 4B). In contrast, the GluSk channel (PDB IDs: 4gx0, 4gx5) demonstrates significant deviations of alpha-carbons from the KcsA template and significant oscillations (Figure 4B), which indicates a mismatch between the sequence alignment and 3D alignment. Another approach to detect π-bulges involves calculation of distances of alpha carbons from the pore axis (Figure 4C). Oscillations in this plot reflect helical structures. At the N-half of the inner helix the plots are congruent, but in positions C-terminal to the π-bulge maxima and minima are shifted one position vs. KcsA. Thus, the two plots unambiguously reveal π-bulges in the inner helices. 

Using this approach, we analyzed cryo-EM structures of eukaryotic sodium and calcium channels. In the NavPaS channel (PDB IDs: 5x0m, 6a91, 6a90, 6a95) [15,16] and chimeric Nav1.7-NavPaS channel (PDB IDs: 6nt4, 6nt3) [17] all the four inner helices have π-bulges at position i14-i16. In the electric eel Nav1.4 channel [18], helix IIS6 has a π-bulge at positions i12-i13 (PDB ID: 5xsy). In the hNav1.4 channel [19], π-bulges significantly reorient residues in the C-terminal parts of helices IS6 and IIIS6 (PDB ID: 6agf). The same pattern of π-bulges in IS6 and IIIS6 is seen in hNav1.2 (PDB ID: 6j8e) [20], hNav1.7 (PDB ID: 6j8i) [21], and rNav1.5 (PDB IDs: 6uz3, 6uz0, 7k18) [22,23]. Figure 5A shows that CA-CB bonds of conserved asparagines in position i20 form two distinct clusters depending on the presence or absence of π-bulges. In S6 segments with the regular alpha-helical structure, the CA-CB bonds of residues N^i20^ are directed toward neighboring inner helices, whereas in S6 segments with the π-bulge they are oriented toward helices S4-S5. 

We analyzed the impact of π-bulges in helices IS6 and IIIS6 on drug binding in the hNav1.4 channel [19] (PDB ID: 6agf). The inner pore of this structure also harbors lipid and steroid detergent molecules. The steroidal detergent binds at the gate region and strongly interacts with numerous pore-lining residues. Particularly, while F^1i30^, F^2i30^, F^3i30^, and F^4i30^ are in matching positions of the aligned repeat sequences (Figure 1), F^2i30^ and F^4i30^ but not Y^1i30^ and F^3i30^ face the pore (Figure 5B) and strongly interact with the steroid Appendix A. Due to the same cause (π-bulges in IS6 and IIIS6, but regular alpha helices IIS6 and IVS6), residues i26 and i22 in different repeats are differently oriented relative to the pore axis, and only in repeats II and IV they interact with the pore-bound ligand. Residues I^3i23^ and I^3i27^ provide large contributions to the ligand-binding energy. Ligand interactions with IS6 are weaker. In the cryo-EM structure of the rNav1.5 channel with flecainide (PDB ID: 6uz0), ligand is shifted toward repeats II and III. Among S6 residues, the largest contributions to the ligand-binding energy are provided by V^2i18^ and F^2i22^ in π-bulged IIS6 and by F^3i16^ and L^3i19^ in IIIS6 with regular alpha-helix (Figure 5C and Appendix A. 

The above comparison shows that due to π-bulges residues in matching positions of the sequence alignment (Figure 1) may have different orientation to the pore axis and different proximity to the pore-bound drugs. Thus, the pattern of π-bulges may critically affect drug interactions with specific channel residues. In contrast, peculiarities of general folding in individual channels have a much milder impact on orientation of drug-sensing residues in the inner helices.

Cryo-EM structures of the rbCav1.1 channel are subdivided into three classes [24] according to the presence of π-bulges in different repeats. In Class I structure [25], S6 helices do not have π-bulges (PDB ID: 5gjv). Complex of rbCav1.1 with nifedipine (PDB ID: 6jp5) represents Class II structures with π-bulges in helices IS6, IIS6, and IIIS6. Structure rbCav1.1 with verapamil (PDB ID: 6jpa), which represents Class III, has π-bulges in IS6 and IIS6 [24]. However, the location of π-bulges is asymmetric. Repeats I and III have π-bulges at position i14-i16, like in Nav1.4 channel, but a π-bulge in repeat II is atypically located at position i21. Cryo-EM structures of rbCav1.1 with various drugs, which are obtained in nanodiscs (PDB IDs 7jpv, 7jpx, 7jpw, 7jpk and 7jpl) [26], belong to Class II, whereas structures of rbCav1.1 without drugs (PDB ID: 3jbr, 6byo) [25,27] lack π-bulges and thus, belong to Class I. As in the case of Nav channels, side-chain orientations of asparagines N^i20^ depend on the presence or absence of π-bulges (Figure 6A).

In the cryo-EM structures of Cav1.1 with DHPs, the ligands bind in the III/IV fenestration. In different Classes, helix IIIS6 at the DHP-binding site has markedly different structures, in particular, different orientations of adjacent methionines M^3i18^ and M^3i19^. In Classes I and III, M^3i18^ is extended to the DHP site, whereas in Class II structure M^3i19^ is extended to the DHP site (Figure 6B). Orientations of F^3i22^ and V^3i23^ are also Class-dependent. Only in Class II structures, the side-chain of F^3i22^ closely approaches the DHP site and V^3i23^ directly faces the inner pore-bound ligands, in particular, diltiazem (PDB ID: 6jpb). 

In the cryo-EM structures of T-type calcium channel hCav3.1 (PDB ID: 6kzo), helix IS6 has a bulge at S^1i15^ [28]. In the hCav3.1 with T-type specific blocker Z944, which stretches from the II/III fenestration to the inner pore (PDB-ID: 6kzp), a π-bulge is also seen at N^2i15^. In both structures, all repeat interfaces are occupied by lipid molecules. Due to the bulge in helix IS6, M^1i18^ does not face the fenestration and does not contribute to binding of a lipid molecule, whereas in the Z944-free channel F^1i16^ significantly contributes to binding of a lipid molecule in fenestration IV/I. Importantly, binding of Z944 correlates with the appearance of a π-bulge at N^2i15^: F^2i19^ in the II/III fenestration strongly interacts with Z944, but in the Z944-free channel F^2i19^ faces away from the fenestration (Figure 6C).

Thus, structures of drug-bound eukaryotic calcium channels illustrate a variety of π-bulge patterns in the inner helices and provide interesting examples of drug-induced π-bulges. In other structures of eukaryotic sodium and calcium channels the π-bulges most frequently appear in repeats I and III. 

### 2.4. Drug Interactions with Lipid and Detergent Molecules 

Lipid and detergent molecules, which are seen in the pore module of many ion channels, may affect the drug-channel interactions. Figure 2D shows that lipid and detergent molecules can occupy the entire space in the inner pore, including the activation gate region, and extend into subunit/repeat interfaces. It is unclear why such molecules are seen only in individual structures. Lipid and detergent molecules can compete with ligands for binding sites in the inner pore. On the other hand, interactions between drug and lipid/detergent molecules can stabilize binding poses of both ligands. For example, in the amlodipine-bound channel Cav1.1 (PDB ID: 7jpx) [26], the drug molecule binds in a repeat interface, whereas its ammonium group reaches into the inner pore and forms a salt bridge with the phosphate group of a lipid molecule (Figure 7A). This interaction, which is four times stronger than any other component of ligand-channel interactions, contributes ~30% to the total binding energy of amlodipine Appendix A.

In the structure of T-type channel hCav3.1 with a T-type selective channel blocker Z944 (PDB ID: 6kzp) a lipid molecule penetrates into the inner pore and forms a salt bridge with the drug molecule (Figure 7B), providing the strongest contribution to the drug-binding energy Appendix A. In the Z944-free structure of Cav3.1 (PDB ID: 6kzo) an aminophospholipid molecule deeply penetrates into the inner pore from the II/III fenestration and forms multiple contacts with the channel residues. In particular, the ammonium group of the lipid is as close as 3.1 Å from the backbone carbonyl of T^2p48^ and 4.4 Å from the backbone carbonyl of K^3p49^. The phospholipid molecule blocks the ion permeation pathway sterically and electrostatically, implying that such a complex would hardly exist at physiological conditions. In the rbCav1.1 channel with verapamil (PDB ID: 6jpa), the classical L-type calcium channel blocker forms a salt bridge with a lipid molecule, which penetrates into the inner pore from fenestration I/II and interacts with a steroidal detergent molecule that extends along the inner pore Appendix A. These interactions largely determine the binding mode of verapamil in the channel [29]. 

### 2.5. Drug Interactions with Permeant Ions

Many inner-pore targeting drugs are organic cations, which may displace permeant cations from their binding sites. On the other hand, electronegative groups of electroneutral and even cationic ligands my favorably interact with permeant ions and thus determine the mechanisms of drug action [30,31,32]. However, many structures lack permeant ions in highly cation-attractive sites of the selectivity-filter region. This may be due to poor resolution of respective structures or peculiarities of protein crystallization/purification protocols. Interrelation between drug binding in the KcsA channel and ionic occupation of the selectivity filter has been demonstrated [33]. Our present computations also predict significant repulsion between the pore-bound cationic ligands of potassium channels and potassium ions in the selectivity filter Appendix A. 

In the CavAb complexes with diltiazem/amlodipine (PDB ID: 6ke5), with Br-verapamil (PDB ID: 5kmh), and in verapamil-bound Cav1.1 (PDB ID: 6jpa), the ligands’ ammonium groups are distant from a calcium ion by 5.5, 6.3, and 6.3 Å, respectively. Calculations predict a strong electrostatic repulsion between the ligands and calcium ions Appendix A. Intensive MC-minimizations of CavAb with Br-verapamil did not reproduce the published drug-binding pose, but the latter was reproduced when a calcium ion at Site 3 was replaced by a water molecule [29]. In the complex of CavAb with efonidipine [11], the carbonyl group of the ligand is 7.7 Å from the calcium ion in the selectivity filter (PDB ID: 6juh). This complex exemplifies an attractive interaction between a drug and a calcium ion Appendix A. Phosphate groups of lipids, which penetrate into the inner pore, favorably interact with the calcium ion in the selectivity filter (PDB IDs: 6jpa, 6kzp and 7jpx). 

## 3. Discussion

The above comparison of published structures of P-loop channels shows greatly diverse patterns of ligand-sensing residues and the pore-targeting drugs, which are scattered in the inner pore from the activation gate to the selectivity-filter and penetrate into fenestrations. Presence of lipid and detergent molecules and a close proximity of binding sites of permeant ions and drugs cast additional uncertainty on mechanisms by which the drugs affect the channel gating and ion permeation. 

Presence of lipid molecules is rather common in cryo-EM structures. Many structures show different transmembrane proteins co-purified with endogenous lipids. An example beyond P-loop channels is a dimeric human GABA_B_ G protein-coupled receptor (PDB ID: 6wiv) in which large endogenous phospholipid molecules are embedded within seven-helical transmembrane domains to maintain their integrity and modulate function of the receptor [34]. 

Our computations suggest that lipid and detergent molecules, besides competing with drugs for binding sites and occluding drug access pathways to the pore, can affect the drug-binding energy and binding poses. It is unclear how relevant are binding poses of drugs complexed with lipid or detergent molecules, which are seen in crystal and cryo-EM structures, to the drugs-binding poses at physiological conditions. Further studies are necessary to address this important problem.

Patterns of drug-sensing residues in eukaryotic calcium and sodium channels are additionally blurred by π-bulges in various positions of different S6 helices, which reorient side chains of ligand-sensing residues and thus affect ligand-channel contacts. The fact that the same channel types have different patterns of π-bulges suggests that the bulges can be dynamic. Moreover, at least two examples demonstrate structural rearrangements due to π-bulges, which are apparently induced by ligand binding (PDB IDs: 6kzo vs. 6kzp and 6jpa vs. 6jpb). Besides affecting ligand-channel interactions, π-bulges may dramatically change interactions between adjacent segments including helices S6, S5, and S4-S5 [35]. As a result, the π-bulges would significantly change the structural stability of the entire pore domain and affect transitions between functional states of the channel.

π-Bulges are seen in other P-loop channels, particularly in two-pore and TRP channels [35], but these channels are not the focus of the present study. Many available structures of TRP channels demonstrate a large diversity of π-bulges in S6 segments. Examples are TRPV1 structures (3j9j, 5is0, 3j5r, 3j5p, 3j5q, and 5irz), TRPM2 structure 6drk, and TRPM4 structures (5wp6, 6bcj, 6bco, 6bqr, and 6bwi), which have π-bulges, and TRPV2 structures (6u84, 6u8a, 5an8, 6bo5, 6bo5, 5hi9, 6bwj, and 6bwm) that lack the bulges. Some structures of TRPV3 channels lack π-bulges in S6 helices (6dvy, 6mhw, 6mho, 6pvl, 6dvw, and 6uw9), whereas other structures (6lgp, 6uw4, 6vpo, and 6mhs) have π-bulges. Such variations suggest that conformational changes in S6 helices can be coupled to appearance and disappearance of π-bulges.

Exact role of such rearrangements remains unclear. Comparison of the TRPM6 channel structures in the open and closed states suggests that the channel opening is accompanied by the α-helix to π-helix transition in the pore-lining helices [36]. Comparison of the TRPV1 and TRPV2 channel structures also suggests that the high-energy π-helical S6 and energetically more favorable α-helical S6 may represent different functional states of the channels [37]. On the other hand, comparison of Class II and Class III Cav1.1 structures shows that the outward motion and axial rotation of helix IIIS6 is accompanied by transition from the π-helix to α-helix [24]. A π-bulge vanishing upon the activation gate widening in the Cav1.1 channel is an opposite example to what is observed in the TRPV6 channel where the π-bulge appears upon the activation gate opening. Thus, there is no strong correlation between presence or absence of π-bulges and dimensions of the activated gate. More likely, π-bulges may facilitate bending of helical segments, which lack glycine residues at position i14, which function as the gating-hinges in KcsA and other potassium channels. 

State-dependent π-bulges and drug-induced π-bulges, which are observed in recent cryo-EM structures, may suggest a novel mechanism by which drugs affect the channel gating. Examples of such drugs are sodium channel activators batrachotoxin and veratridine, and DHP agonists and antagonists of L-type calcium channels. These modulators affect channel gating by increasing and decreasing, respectively, probabilities of the open channel conformations [38,39,40,41,42]. Atomic-scale structures of sodium channels with agonists are still lacking. Comparison of the Cav1.1 structures with the DHP agonist (PDB ID: 7jpk) and antagonist (PDB ID 7jpw) bound in the III/IV fenestration [26] shows only small changes in the activation gate region. However, a possibility that drug-induced π-bulges could reorient S6 residues and thus affect stabilities of the open- and closed gate conformations deserves further experimental and theoretical studies.

In summary, the impressive progress in structural studies of P-loop channels and their complexes with drugs explained numerous data accumulated in decades of intensive experimental studies, but highlighted new problems that should be addressed in further structural and functional studies. In particular, the origin of π-bulges and their impact on channel gating and drug binding appears as a novel important problem in structural pharmacology of P-loop ion channels. 

## 4. Methods

We used a residue labeling scheme elaborated in our previous studies [3]. In eukaryotic sodium channels repeat domains I to IV are arranged clockwise when viewed from the extracellular side. Similar arrangement is proposed for homo- and hetero-tetrameric channels. Initial sequence alignment was performed by CrustalW and manually corrected to ensure the same orientation of helix residues in homologous positions. In particular, the GMG motif in the P-loop of TRP channels was shifted relative to the GYG motif in potassium channels [43] and insertions/deletions were introduced in the P-loops of eukaryotic sodium channels aligned with bacterial sodium channel NavAb [44,45].

The structures were 3D-aligned against the open potassium channel Kv1.2/Kv2.1 (PDB ID: 2r9r), the first eukaryotic P-loop channel whose crystal structure was obtained with a resolution below 2.5 Å [46]. Initially, we re-oriented the Kv1.2/Kv2.1 structure so that the pore axis coincided with the *z*-axis, the tyrosine backbone carbons in the fingerprint selectivity-filter GYG motif laid in plane *xOz*, and axis *Ox* directed toward the GYG tyrosine in subunit III. Other structures were 3D aligned with the Kv1.2/Kv2.1 structure by minimizing RMS deviations of alpha carbons in positions p38 – p47 of four P1 helices, which are most 3D conserved elements in P-loop channels [45].

Energetic characteristics of ligand-channel complexes were calculated with the AMBER force field [47] where most of H-bonding energy is a part of non-bonded energy. Hydration (solvation) energy was calculated with implicit-solvent method [48]. Electrostatic energy was calculated with distance-dependent function *ε = 2r*, where *r* is the distance between atoms. The atomic charges in ligands have been calculated by the semi-empirical method AM1 [49] using MOPAC. Structures were optimized by Monte Carlo (MC) minimizations [50] using the ZMM program [51]. Starting conformations of side chains, which are unresolved in the PDB structures, were assigned with SCWRL4 [52]. To ensure folding similarity of the MC-minimized and PDB-deposited structures, C^α^ atoms of the models were constrained to experimental positions by pins. A pin is a flat-bottom parabolic penalty function that imposes the energy of 10 kcal mol^-1^ Å^-1^ if deviation of an atom from its experimental position exceeds 1 Å. Pin constraints were also applied for ions and all heavy atoms of ligands. The energy of MC-minimized structures was partitioned by channel residues (see Appendix A). 

## Figures and Tables

**Figure 1 ijms-22-08143-f001:**
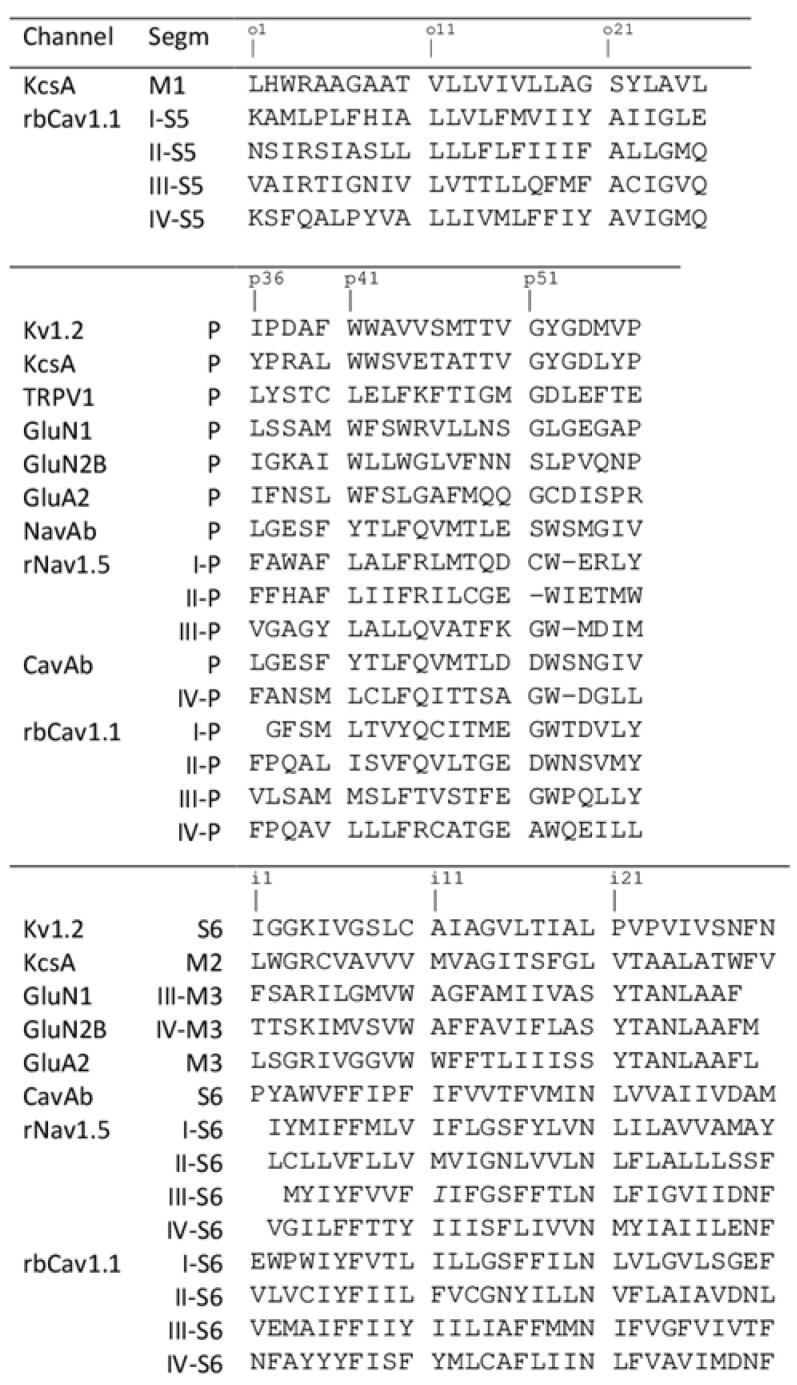
Sequence alignment of helices in the pore domain of P-loop channels. Universal labels of residues are shown above the sequences.

**Figure 2 ijms-22-08143-f002:**
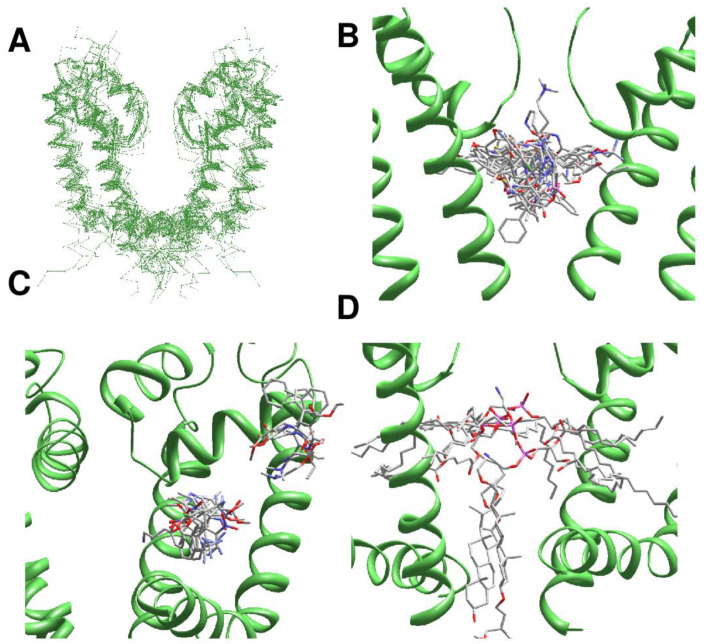
**Ligands in the pore domain of P-loop channels.** Carbon, oxygen and nitrogen atoms of ligands are gray, red and blue, respectively. (**A**), C^α^ tracing of the pore domain in 25 drug-bound P-loop channels. For clarity, only two subunits/repeats without drugs are shown. The structures are 3D-aligned as described in Methods. (**B**), Superimposition of ligand-channel complexes with ligands in the inner pore. Only two subunits of the KcsA channels are shown for clarity. Ligand molecules fill the entire central cavity. (**C**), Superimposition of channel complexes with ligands bound in subunit/repeat interfaces (between P1 helix and two inner helices). Most ligands are close to the pore. Exceptions are DHP drugs in CavAb, which are far from the pore axis. (**D**), Lipid and detergent molecules bind in the inner-pore central cavity, fenestrations, or in the activation gate region.

**Figure 3 ijms-22-08143-f003:**
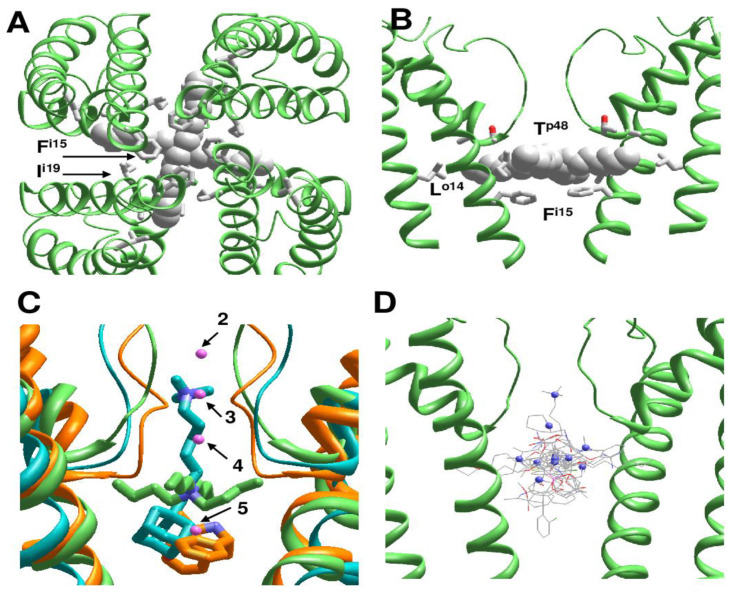
**KcsA pore domain (green helices) with superimposed ligands from KcsA, GluR, NavAb, and CavAb channels.** (**A**,**B**), Intracellular and intra-membrane views of the KcsA complex with tetradecylammonium (PDB ID: 2w0f). The ammonium nitrogen is at the focus of P-helices. Long decyl chains penetrate into subunit interfaces and reach the outer helices. Main channel contributors to the ligand-binding energy are shown by sticks. (**C**), 3D-aligned structures of TBA-bound KcsA (PDB ID: 2hvj) (green backbone and ligand carbons), MK801-bound NMDAR channel (PDB ID: 5un1) (orange backbone and ligand carbons), and IEM-1460 bound AMPAR channel (PDB ID: 6dm0) (cyan backbone and ligand carbons). The central cavity of the channels accommodates bulky moieties of TBA, MK801, and IEM-1460. Magenta spheres show potassium ions in Cd^2+^-bound KcsA (PDB ID: 3stl). The terminal ammonium group of IEM-1460 penetrates into the selectivity filter and binds near Site 3 of potassium ion; the second ammonium nitrogen binds close to the position of TBA ammonium nitrogen in KcsA. The ammonium group of MK801 binds in the central cavity, close to Site 5 of potassium ion KcsA. (**D**), Ammonium nitrogen atoms of cationic drugs (blue spheres) scatter over the central cavity.

**Figure 4 ijms-22-08143-f004:**
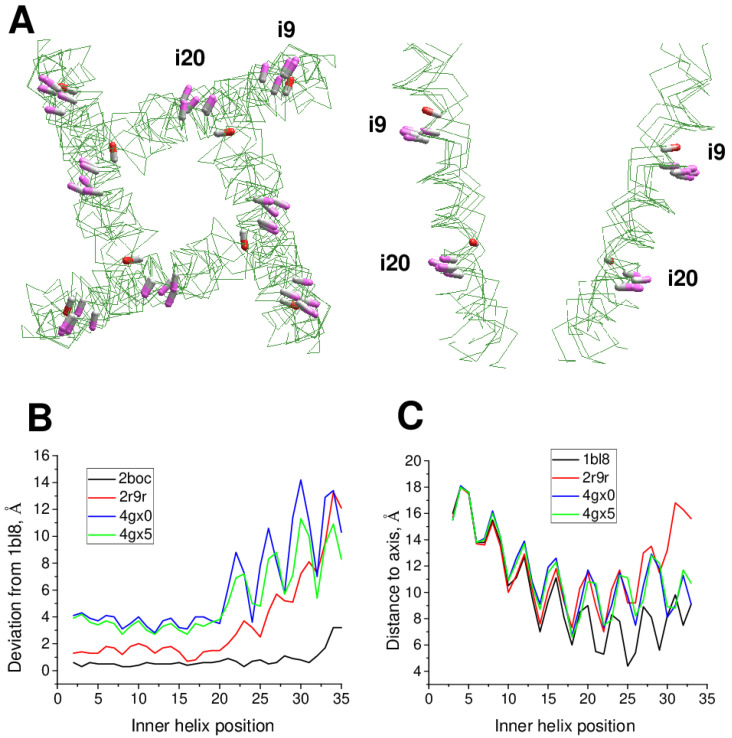
**π-Bugles in the inner helices of P-loop channels.** (**A**), Intracellular and membrane vies at superimposition of KcsA (PDB ID: 1bl8), Kv1.2/Kv2.1 (PDB ID: 2r9r), AMPAR (PDB ID: 6dm0), NavAb (PDB ID: 3rvy), NavMs (PDB ID: 6yz0), and GsuK (PDB ID: 4gx5). CA-CB bonds in positions i9 and i20 are shown as sticks. Upstream from the π-bulge (position i9), orientations of CA-CB bonds (gray CA, pink CB) are similar, while downstream the bulge (i20 position), orientation of CA-CB bonds in GsuK (gray CA, red CB) is different from that in other channels. (**B**), Deviations of alpha carbons in representative structures of P-loop channels from sequentially matching positions in the KcsA structure (PDB ID: 1bl8). Another KcsA structure (PDB ID: 2boc) has small deviations. For the open-gate Kv1.2/Kv2.1 channel, the deviations are large at the C-terminal part, but the deviation plot remains rather smooth. For the GsuK structures (PDB IDs: 4gx0, 4gx5), with a bulge in the middle of the inner helix, large oscillations are seen at the plots. (**C**), Distances of alpha-carbons in the inner helices from the pore axis. The bulge in the GsuK structures causes a shift of maxima and minima in the distance plot.

**Figure 5 ijms-22-08143-f005:**
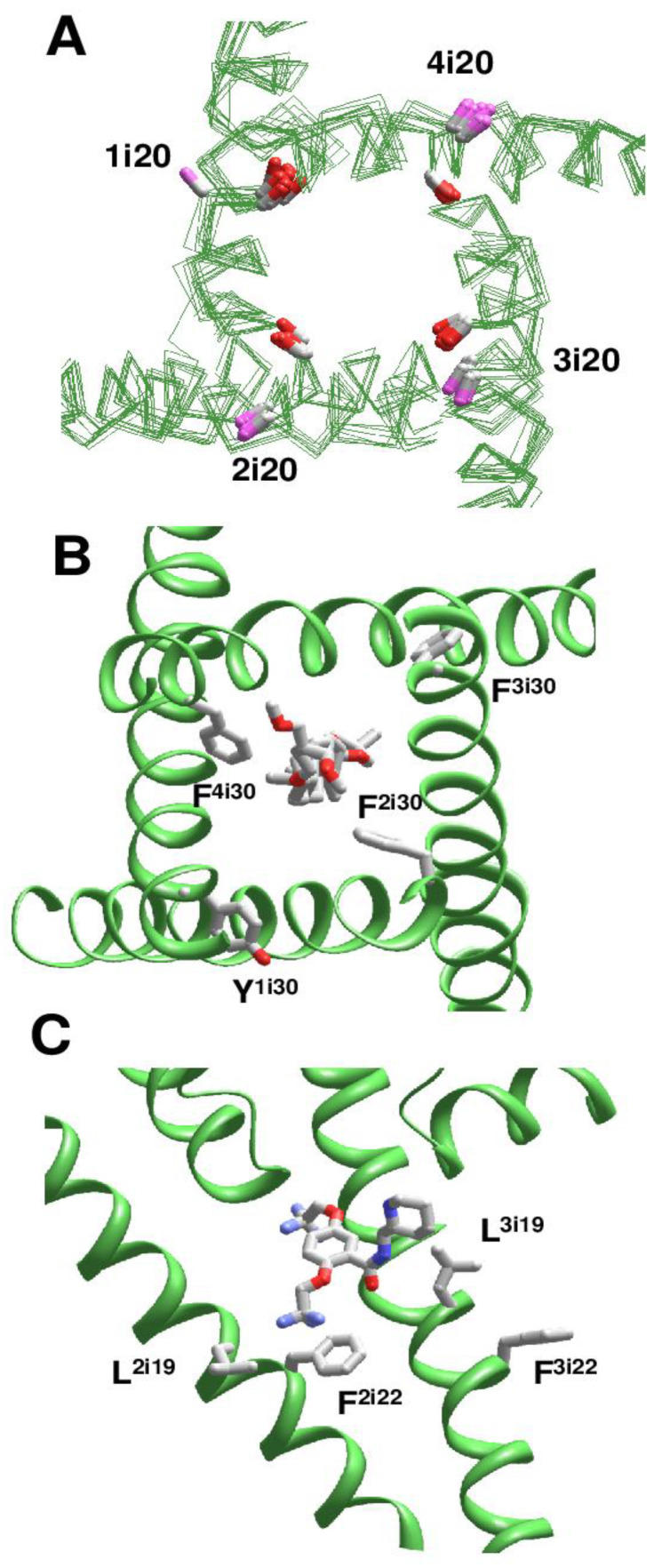
**π-Bulges affect patterns of ligand-sensing residues in eukaryotic Nav channels.** (**A**), N^i20^_C^α^-C^β^ bonds in regular alpha-helices and π-bulged helices are shown with magenta and red CB atoms, respectively. **B**, A steroidal molecule in hNav1.4 (PDB ID: 6afg). F^2i30^ and F^4i30^ face the pore and interact with the steroid, while residues in sequentially matching positions, F^3i30^ and Y^1i30^ (Figure 1) face away from the pore. ***C***, Flecainide in rNav1.5 (PDB ID: 6uz0). Residues F^2i22^ and L^3i19^ are oriented toward the pore-bound ligand, whereas F^3i22^ and L^2i19^ are oriented away from the pore.

**Figure 6 ijms-22-08143-f006:**
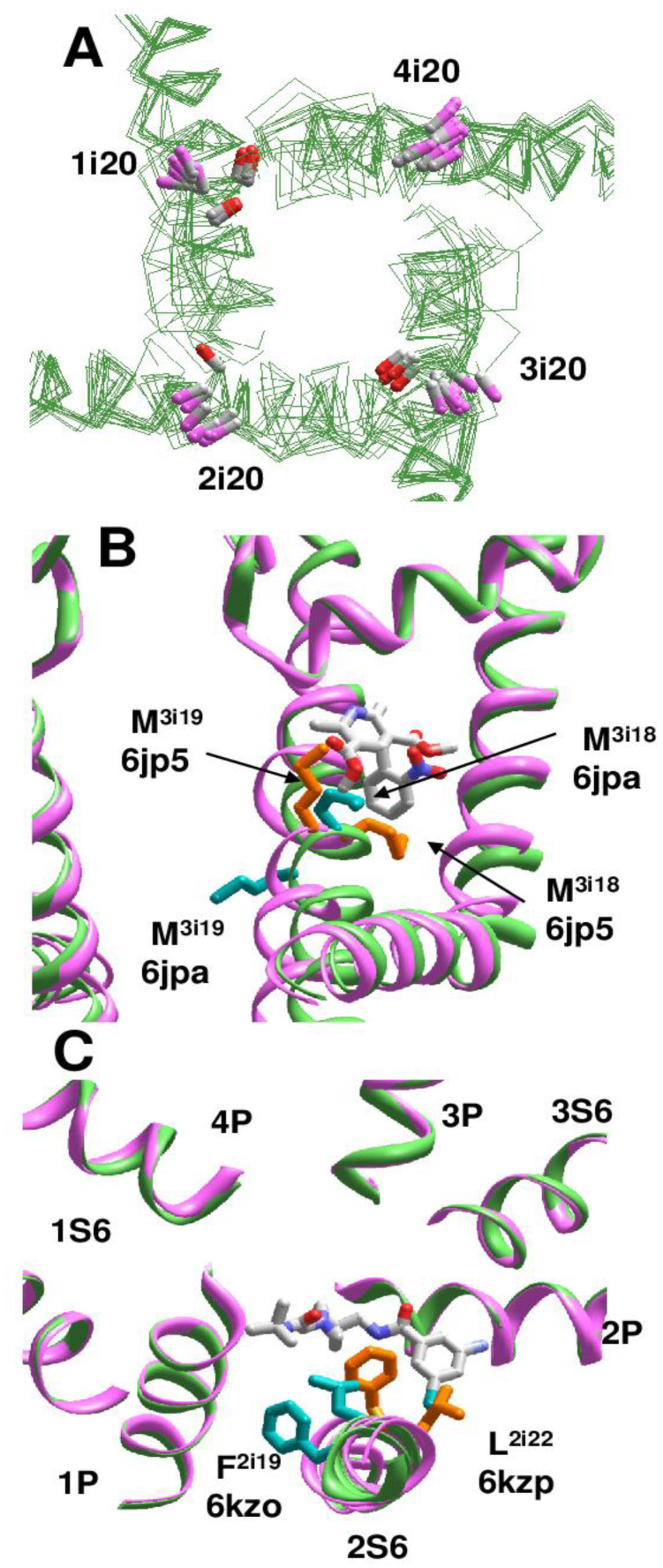
**π-Bulges affect patterns of ligand-sensing residues in Cav1.1 channel.** (**A**), Diverse orientation of asparagines N^i20^. Orientation of N^i20^_C^α^-C^β^ bonds in regular and π-bulged helices are shown with magenta and red CB atoms, respectively. (**B**)***,*** Different orientation of adjacent M^3i18^ and M^3i19^ in Cav1.1 Class II structure (PDB ID: 6jp5, magenta backbone, orange sidechains) and Class III structure (PDB ID: 6jpa; green backbone, cyan sidechains). M^3i18^ faces the DHP site in Class II structure, whereas M^3i19^ faces the DHP site in Class III structure. **C,** Superimposition of Cav3.1 structures with Z944 (6kzp) and without Z944 (6kzo). Residues in matching positions 2i19 and 2i22 differently interact with the ligand since the π-bulge is present only in 6kzp.

**Figure 7 ijms-22-08143-f007:**
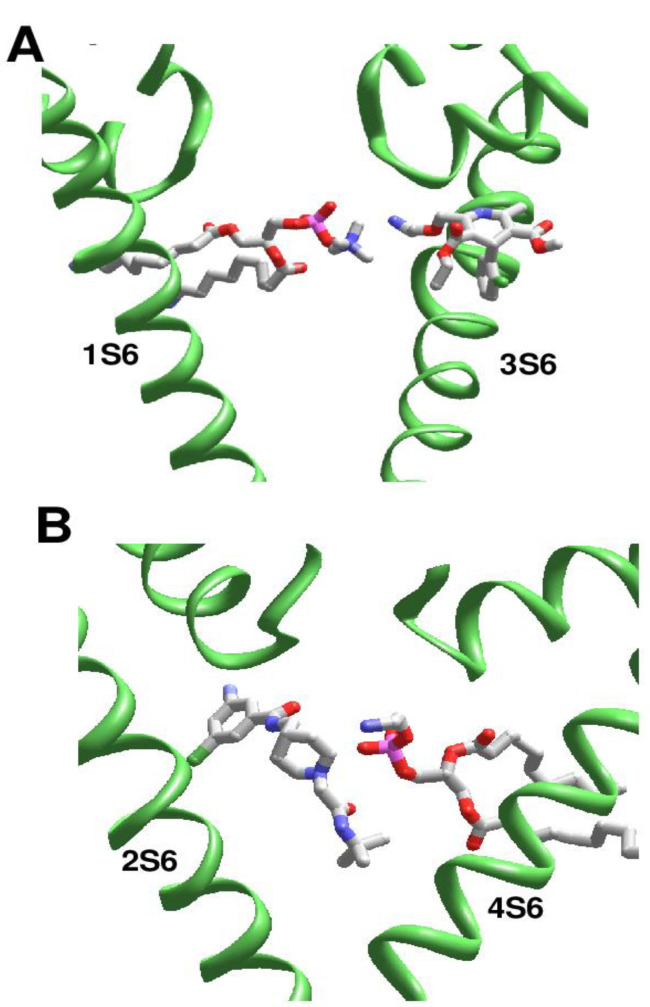
**Examples of ligand interactions with the lipid molecules in the pore.** (**A**), Amlodipine in Cav1.1 (PDB ID: 7jpx). The ammonium group approaches the outer pore and forms a salt bridge with a phospholipid molecule. (**B**), Z944 and lipid molecules fill up the cavity in Cav3.1 (PDB ID: 6kzp). The ligand amino group and phosphate group of the lipid (pink phosphorus) are salt-bridged.

## Data Availability

The 3D-aligned structures are available on request.

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
