# Peer review of "Computational Analysis of the Crystal and Cryo-EM Structures of P-Loop Channels with Drugs"

_ijms, 2021, doi:10.3390/ijms22158143_

Round 1

Reviewer 1 Report

In the present study, the authors 3D-aligned structures of different P-loop channels, designated residues with universal residue labels, and estimated energetics of ligand-channel interactions, and then described common and specific features of ligand-channel complexes.  This study provides novel insight into these structure-channel relationships and allows a framework for continued investigation and development of novel pharmacotherapeutics. 

Author Response

Dear Reviewer,

Thank you very much for reviewing our manuscript.  In the revised version we introduced some changes according to comments of another Reviewer.

Sincerely,

Denis Tikhonov and Boris Zhorov

Reviewer 2 Report

The authors studied experimentally solved P-loop family channels and they detaield interactions with their pharmacological molecules. The study here is very original and the results are significant. I highly recommend to publish their paper after a few minor points addressed. 

  1. Some of the PDB codes are typos I believe. For example, page 6 line 144, 2hvj not 2hjv. same page, line 155, 2w0f not 2wof, line 215 etc. 
  2. Potassium channels are clearly dicussed the pi-bulge effect, however, Cav, Nav and TRP familly are far more complicated. If the authors could address more on Cav and Nav, it would be better.
  3. The interaction between ligand and lipid molecules are very common in cryo-EM structures. Some endougenous lipids are co-purified with the protein and because of their tide binding to the protein, their experimental density is very clear. So when drugs or ligands are added, their interactions are clearly visible in the raw data, so pdb files from cryo-EM experiments often modeled with lipids or detergents. Here is nice example Nature 584 (7820), 304-309 if the authors are interested.  

Author Response

Dear Reviewer,

Thank you very much for reviewing our manuscript.  In the revised version we introduced several changes according to your comments. Below please find point-to-point answers to these comments.

Comment 1. Some of the PDB codes are typos I believe. For example, page 6 line 144, 2hvj not 2hjv. same page, line 155, 2w0f not 2wof, line 215 etc. 

Response. Thank you for spotting the typos. PDB codes are corrected.

Comment 2. Potassium channels are clearly discussed the pi-bulge effect, however, Cav, Nav and TRP families are far more complicated. If the authors could address more on Cav and Nav, it would be better.

Response. At present, available structures of calcium and sodium channels are limited and we cannot to extend the consideration of pi-bulge effects . Many structures of TPR channels are available, but these channels are not in the focus of our present study. We added a paragraph in the discussion, which demonstrates large variations of these structures regarding pi-bulges between TRP subtypes and within TRPV3 family.

Comment 3. The interaction between ligand and lipid molecules are very common in cryo-EM structures. Some endogenous lipids are co-purified with the protein and because of their tide binding to the protein, their experimental density is very clear. So when drugs or ligands are added, their interactions are clearly visible in the raw data, so pdb files from cryo-EM experiments often modeled with lipids or detergents. Here is nice example Nature 584 (7820), 304-309 if the authors are interested.  

Response.  Thank you for bringing our attention to this interesting structure. We added a brief description of the structure as an example of lipid roles in structures beyond the family of P-loop channels. We also mentioned that while some drug-lipid interactions seem obvious upon visual inspection of 3D structures, our computations demonstrate that such interactions can be very strong and therefore they may significantly affect the drug binding poses and patterns of drug interactions with the channels.

Sincerely,

Denis Tikhonov and Boris Zhorov